# Zoonotic Disease Risks of Live Export of Cattle and Sheep, with a Focus on Australian Shipments to Asia and the Middle East

**DOI:** 10.3390/ani12233425

**Published:** 2022-12-05

**Authors:** Clive J. C. Phillips

**Affiliations:** 1Institute of Veterinary Medicine and Animal Sciences, Estonian University of Life Sciences, Kreutzwaldi 1, 51014 Tartu, Estonia; clive.phillips@curtin.edu.au; 2Curtin University Sustainability Policy (CUSP) Institute, Curtin University, Kent St., Bentley, WA 6102, Australia

**Keywords:** animal welfare, Australia, cattle, live export, sheep, zoonotic disease

## Abstract

**Simple Summary:**

Most human disease emanates from animals, and there is a significant risk of zoonotic diseases being transmitted from livestock that travel long distances between countries as part of the live export trade. Both cattle and sheep are subject to many stressors when transported long distances by ship, including starvation, ship motion, ammonia, heat stress, high stocking density, an unhygienic environment and mixing of animals. These are believed to deplete immune system responses, fostering disease transmission between livestock. Several zoonotic diseases commonly occur in livestock exported from Australia, and this sometimes triggers rejection by importing countries. Trade restrictions have been imposed in the past by both importing and exporting countries as a means of controlling disease spread. It is concluded that the live export trade in cattle and sheep poses a significant risk of spreading zoonotic diseases.

**Abstract:**

The growing human and livestock populations in the world today and increased international transport of livestock is increasing the risk of both emerging and endemic zoonotic diseases. This review focuses on the potential for the live export trade to transmit zoonotic diseases. Both cattle and sheep are exposed to major stresses during the transport process, which are described, together with the impact of these stresses on the immune function of transported animals. Heat stress, overcrowding, inanition, ship and vehicle motion and accumulation of noxious gases are analysed for their ability to potentiate infectious diseases. The major zoonoses are described: pustular dermatitis, pneumonia, salmonellosis, as well as some common conditions, such as conjunctivitis, with specific reference to stressors associated with each disorder. Historical precedents exist for restriction of the trade based on disease risks. Finally, the economic and regulatory frameworks are considered to evaluate ways in which the spread of zoonotic diseases can be controlled.

## 1. Introduction

Zoonotic diseases are the major cause of morbidity in humans, responsible for nearly two thirds of infectious disease, about a billion cases of human illness annually [1]. Much has been written about the implications of the live trade in wildlife for zoonotic disease transmission, but rarely have the implications of the growing trade in livestock been considered [2]. However, it *is* known that the spread of foot-and-mouth disease in Africa, Europe, the Middle East and Asia is linked to livestock trade and that the transmission of BSE into several countries was linked to trade in infected cattle [2]. Modern rapid transportation systems and intensive, high density management of livestock farming systems encourage emerging and existing pathogens to multiply and risk human health globally [1]. Over the last century, trade liberalization, reduction in transport costs and growing consumer affluence have encouraged worldwide expansion of the live animal trade [3]. In the last 30 years, Australia has operated a major export industry in livestock, principally cattle and sheep, sent mostly by ship to South-East Asia and the Middle East, respectively. Towards the end of the last century, the high prices paid for live animals led to the demise of abattoirs in the north of Australia, restricting the possibility of home processing of the cattle produced in Australia [4]. 

The intimate connection between human and animal diseases, recognised in the One Health concept [5], suggests that better awareness of the risks posed by the growing trade in live animals is warranted. With most human diseases originating in domestic animals or wildlife [6], and human and domestic animal numbers concurrently increasing rapidly, the risk posed by new and existing zoonoses is concomitantly increasing [1]. The susceptibility of the human host to novel zoonoses from livestock is also increasing, because as the populations of both grow, high density living, for both humans and livestock, are adopted as the only solution to the need for efficient accommodation systems and, in the case of livestock, the need to produce cheap food. This increases the disease transmission risk by virtue of the reduced inter-individual distance [7] and, where it is associated with more stressful living conditions, depletes the immunocompetence that enables them to deal with the disease challenges [6]. This is nature’s way of steering animal populations towards more sustainable living. In the 21st century, additional pressure is being placed upon human and livestock populations by global climate change, with more anticipated cross-species disease transmission [8]. Despite these dynamic processes, it is also recognised that there are many zoonoses that are enzootic in nature, i.e., stable and established in the animal population, which may transmit to humans with no subsequent human to human transmission [1]. 

In Australia, the 615,000 cattle exported annually are economically more significant than the almost 500,000 sheep involved in live export, worth AUD billion $1 and 0.085, respectively [9]. The many countries to which Australia exports livestock (termed an outdegree country in network analysis [2]) provide significant potential to transmit zoonoses to other markets. The extent of transitivity, i.e., where countries with which Australia trades also conduct trade with each other [2], has not been studied. Australia also has several triadic relationships, i.e., countries to which Australia exports to that go on to sell the livestock to other countries, which may be used to overcome trade restrictions. The Australian markets have been growing recently [10], but also changing from one dominated by cattle exports to Indonesia to a more diversified cattle market with growth in Vietnam and China, and to a lesser extent Malaysia and Russia [11]. Cattle are also exported to Japan, the Middle East, North Africa, and countries surrounding the Black Sea. Health certification standards vary according to the importing country, e.g. a particularly high standard of health certification is required for cattle exported to Japan, including 21 d isolation, vaccination for a range of the most serious diseases and negative test responses to assays for a range of other major diseases [6]. On board, the most important causes of cattle mortality, which is less than that for sheep, are heat stress, trauma and respiratory diseases [7]. 

Despite worldwide growth in the live animal trade, after significant growth at the end of the 20th century, Australian sheep exports have declined steadily from 4 million in 2003 to 0.5 million today [11]. This reduction is mainly because of prolonged droughts in Australia and, more recently, concerns about the continuity of supply in the face of government restrictions on exports, in particular, the ban on exports during the Australian winter, which arrive at importing countries in the Northern Hemisphere summer [12]. Australian sheep are exported to a variety of countries around the world, including Kuwait, United Arab Emirates, Jordan, Oman, Bahrain, Qatar, Egypt, Israel, Lebanon, territories administered by Palestine, Ukraine, Malaysia, Singapore, Mauritius, New Zealand, Vanuatu, Brunei, China, Japan, USA, Mexico, Argentina, Chile and the Philippines. The countries in the Gulf Cooperation Council (GCC), i.e., the Arab states of the Persian Gulf, are the most common destination, particularly Kuwait, Qatar and UAE, as well as Jordan. Imports increase in advance of the festival of Eid al-Adha [13], when large numbers of sheep are sacrificed in remembrance of God saving the son of Ibrahim (Abraham, to Christians) and replacing him with a sheep [14]. The meat is shared with family, neighbours and poor people, which adds a social benefit [15]. Sheep destined for GCC may be offloaded at several ports in the Gulf [16]. The risk of heat stress in the Persian Gulf during summer is very high [16], particularly since the festival, which advances by approximately 11 days each year, now occurs soon after midsummer in the Middle East [17]. The Australian government has restricted sheep shipments during the middle of the Australian winter because of the high risk of heat stress to sheep in the Persian Gulf [12]. 

Australia has a large feral and domestic goat population [18]. In the past, goats were exported by sea and, although there have been attempts to recommence this trade [19], permits for export by sea are generally not issued by the Australian Quarantine and Inspection Service. Goats are therefore mainly exported by air from Australia to Malaysia and Singapore.

The aim of this study was to review the literature relevant to zoonotic diseases that might be transmitted during the live export process, in order to consider the risk of transmission of diseases from livestock to humans. The search strategy for this review was to primarily use the Web of Science for relevant scientific articles and, where necessary, to search for popular material using internet-based search engines. Key terms were zoonotic or zoonosis/zoonoses and livestock (cattle and sheep), together with individual diseases and transport stressors of relevance. High-quality references were chosen that demonstrated rigorous methods in their research. Non-peer-reviewed material, in particular that published by credible sources, such as the WHO, OIE, FAO/WOHA, were included as additional information for the reader. Industry sources, e.g., Livecorp, were cited where no peer-reviewed literature was available. The author also has firsthand experience of travelling with live export shipments from Australia. 

## 2. The Supply Chain

In this section, the long process of preparation for export is described, in order for the reader to understand the considerable stress on animals before the ship journey begins. Most cattle are sourced from the north of Australia, especially the Northern Territory and Western Australia [20]. Almost all the sheep for the Middle East trade are sourced from Western Australia [21] and, therefore, travel a maximum of about 12 h to the depot. 

Cattle are usually initially mustered with helicopters and/or aeroplanes [22], after which a ground-based team using vehicles and motorbikes direct the cattle towards handling yards. From the yards, they are taken by truck to an assembly depot [23], where they are mixed with cattle from other properties [24]. Sheep are mostly mustered in the paddock by farm workers on motorbikes, with the aid of dogs. Mustering is acknowledged to create stress [25], hence both cattle and sheep spend several hours in yards, often being held overnight, so that they can begin their onward journey the next day [26]. This helps them to cool down and reduces their stress level before departing by truck for the pre-embarkation assembly depot. Transport is usually by one-tiered (for cattle) or two-tiered (for sheep) naturally ventilated trucks with one to three trailers. Capacity of the trucks is typically 50–100 cattle or 700–900 sheep, depending on the animals’ live weight and, in the case of sheep, wool length [11]. They are stocked at high density and the floor is ridged so that they do not lie down [23]. If they lie down, they may be unable to get up again if other animals are standing over them.

After being conveyed by the transport company to the assembly depot, they are supposed to remain there for at least 5 days to adjust to, firstly, close confinement with sheep from other properties and, secondly, the change in diet from pasture to pellets [24]. This will change their microbiome, which, depending on exposure, may in turn increase the abundance of human pathogens, such as *E. coli* O157 [27]. On the day before, or the day of, departure, they are taken by trucks from the depot to the port, where they are again mixed with other stock. During the loading process, they are inspected by an Australian Quarantine Inspection Service officer, who is meant to detect signs of ill health, in particular, lameness, pink eye, nasal discharge and body lesions, any of which should make them unfit to load. Subclinical disease, however, and some clinical disease will not be detected. After inspection, they enter the ship, where they are directed into their pen via a series of ramps. 

On board the ship, the animals are directed into pens, holding tens of cattle to a few hundred sheep at stocking densities similar to those used for transporting livestock by truck [24]. The ship journey that follows usually lasts for between 1 and 5 weeks, depending on the destination [12,21,28]. Daily management is conducted by the crew, under direction from one or two stock people and sometimes an accompanying veterinarian. A description of the stress imposed by their conditions is provided below [29]. 

Following disembarkation, livestock (particularly cattle) are either delivered directly to a feedlot or they are loaded into vehicles to transport them to an abattoir. The transport systems are usually not as sophisticated as those used in Australia to take them to the ship, with some animals being loaded onto/into cars and others into small trucks. Road quality is variable and transport distances can be long—for example, several thousand kilometres in China. 

After a period in a feedlot for further growth, animals are usually delivered, either alive or as carcasses, to wholesalers who send the meat to retailers operating in wet markets [30]. Wet markets are widespread in Asia and the Middle East, where meat has traditionally been purchased fresh each day [31]. So far, the availability of refrigerated storage has been limited, but this may change. The United Nations (UN) acting Head of Biodiversity Patrick Greenfield has called for ‘countries (to)… move to prevent future pandemics by banning wet markets that sell live and dead animals for human consumption’ but he cautioned against unintended consequences [32]. For example, this might encourage a black market to develop, which could only be controlled if public opinion were to galvanise against purchasing meat from underground sources [33]. To date, there are no legal restrictions of wet markets. In the case of Australia’s live sheep exports, Saudi Arabia was traditionally the biggest market, but in 2012 it refused Australian live sheep exports because the (live) Export Supply Chain Assurance Scheme insisted upon by Australia challenged Saudi’s sovereignty over imports. However, in 2021 sheep exports were accepted provided they had been vaccinated against pustular dermatitis [34]. 

OIE guidelines for transport of livestock are applied with varying degrees of conformity, producing concerns over potential transmission of zoonoses in recipient countries [35]. Alternatives to Australia’s live sheep exports for Saudi Arabia are live sheep from the Horn of Africa and lamb meat products exported by New Zealand, Pakistan, Sudan and other African countries [36]. Abattoir workers are particularly likely to be exposed to zoonoses around the time of the Hajj, with approximately 15,000 workers normally brought in for this purpose from Syria, Egypt and Turkey [37,38]. A study conducted in 2013 found a relatively high prevalence of Brucella exposure (13%) in these abattoir workers [37]. During the COVID-19 pandemic, the importation of livestock was reduced because the Hajj was cancelled, reducing demands for livestock products by pilgrims [39]. Saudi has a young, rapidly growing population that may be persuaded to redirect purchases to imported meat rather than live animals. Road transport networks within the country are well-developed, rail freight is improving and there is generally good infrastructure [40]. 

Competition for Australian cattle exports comes mainly from Brazil, which now exports to Indonesia, Middle East and India. India mainly exports buffalo meat, particularly to Indonesia [41]. 

## 3. Stresses on Board and Disease Risks

There are several diseases that pose a risk to exported livestock, which, when infected, may transmit the disease to the local livestock population and, if they are zoonotic, to the human population [42]. The risk of disease is likely to be increased by the unhealthy and stressful conditions on Australian live export ships, as well as the stress over several weeks, and in some cases months, as the animals make the long journey from their paddock in Australia to the slaughterhouse in an importing country [43]. 

### 3.1. Stressors during Live Export and Their Impact on the Immune System

Stress impacts on immunocompetence through the hypothalamic–pituitary–adrenal axis and the sympathetic–adrenal–medullary axis [44]. This diminishes vaccine responses and enhances bacterial and viral pathogenesis. 

### 3.2. Starvation

Temporary starvation is a common response to long-distance transport by sea, with sheep often not feeding for the period that they spend in an assembly feedlot (usually about 5 days) before loading [45,46] and in the early stages of the sea voyage [47]. As well, many ships carry inadequate feed supplies and have to ration livestock, sometimes reducing feed provision by as much as 36% of the recommended allowances [48]. This is likely to increase the proportion of human pathogens, such as *E. coli* O157, in the animals’ microbiome [27]. The stress induced by long-distance transport reduces the richness of the intestinal microbiota, which in healthy animals are able to resist colonisation of the intestines by pathogens through their production of lactic acid and short-chain fatty acids ([49]. Such immunocompromise during transport is well established for the gut microbiome, but it is less well understood for the respiratory tract [50]. 

Starvation in livestock induces the release of ketone bodies, including β-hydroxybutyrate [45,51,52], which compromise the mucosal defences in the nasopharynx, reducing the diversity of the microbiota and, in the intestines, reducing the number of benign *Bifidobacterium* and *Lactobacillus*, allowing colonisation by pathogens [53]. As well as effects on mucosal microbiota, long-distance transport also directly reduces immune system responses, mainly T rather than B lymphocytes [54]. Some nutritional strategies are available to limit these effects: in Turkish sheep transported long distance, vitamin C supplementation increased immunocompetence [54]. No benefit of providing supplementary vitamin E on stress responses of transported ewes has been detected [55]. 

Sheep willingness to tolerate starvation appears to be influenced by the time of year. Sheep use a declining daylength as a cue to start breeding, somewhat earlier in rams than ewes, which increases appetite [56]. Sheep mortality rates during voyages to the Middle East are higher on shipments leaving Australia in the second half of the year, when daylength is increasing [45]. Inappetence was the primary cause of sheep mortality in August shipments [57], associated with inadequate fat mobilization, following good grass availability in south-west Australia at this time [45]. In a later analysis of 417 voyages, there was a higher mortality rate when ships left Australia in the Southern Hemisphere winter or spring, when daylength was short but increasing, compared with summer or autumn, when it was long but decreasing [58]. Departure in the Southern Hemisphere autumn, particularly, had a low sheep mortality rate [58]. This is probably more due to circannual nutritional changes in Australian sheep than specific effects of daylength on metabolism. Sheep in Australia start mobilising fat tissues in autumn after summer drought and the associated nutrient shortages [59,60]. Evidence of fat mobilisation has been provided [23], in the form of increased plasma non-esterified fatty acids, beta hydroxybutyrate and glycerol in sheep at this time. Carnovale and Phillips [16] noted particularly low mortality in voyages leaving Australia in May, during short and declining daylight, compared to the rest of the year. The sudden exposure to the continuous light on the ship, left on to reduce accidents, may make these sheep perceive that they have transitioned into summer, increasing appetite so that they can store fat for the winter. This may make the sheep less tolerant of starvation and more likely to survive. 

### 3.3. Ship Motion

Short-term studies have demonstrated that floor motion, similar to that experienced on ships, causes stress to sheep, with the heave motion causing more stress than other movements, such as roll and pitch [61]. These studies were conducted with repeated presentations of short periods of motion, 30 min to 1 hour; it is not known if a continuous period of motion increases or decreases the observed stress responses. 

Roll motion requires sheep to regularly engage in stepping behaviour as a coping mechanism; however, evidence from laterality studies and heart rate responses suggests that it is also stressful [62]. The more irregular the motion is, the more stressful it is for the animals [63]. 

Simulated ship motion has adverse effects on feeding behaviour and balance of sheep, but these can be attenuated by administration of an antiemetic, suggesting that seasickness is experienced [64]. Motion sickness is not proven in livestock, but is likely, because of the expectation that a regular challenge to the vestibular system and potential losses of balance accompany ship movement [65]. It is not necessarily associated with vomiting, because of the configuration of the abomasum and rumen, but sheep subjected to motion stress appear to feel a malaise [64]. Motion sickness is associated with elevated levels of tumour necrosis factor alpha, which is one of the primary regulators of immune cells [66]. 

To reduce the stress of motion for the animals, a catamaran hull formation, rather than the existing monohulls, resists roll motion more effectively and allows more transverse movement within the vessel without compromising safety [67]. Furthermore, at high speeds it offers less resistance and is, therefore, inherently more efficient. 

### 3.4. Ammonia

Ammonia typically averages 17–19 ppm over the course of a sheep voyage from Australia to the Middle East, but individual locations may reach 59 ppm or more [68]. High levels on board cause stress to sheep, reduce feed intake [69,70,71], and increase the incidence of conjunctivitis [70] and pulmonary inflammation [71]. Pulmonary inflammation is likely to increase susceptibility to serious bacterial pneumonia [72]. There is also circumstantial evidence that the buccal cavity is inflamed by ammonia, which may predispose sheep to pustular dermatitis [71]. 

### 3.5. Heat Stress

Sheep and cattle travelling northwards from Australia are subjected to increasing temperatures, exacerbated by their inability to cool down at night [16]. Evidence of heat stress is widespread, with 50% of voyages to two of the four Persian Gulf ports experiencing temperatures sufficient in summer to cause heat stress [16]. Cattle respond by panting, orientating their heads to the ground, and raising their tails, attempting to lose heat [73]. Heat stress has similar clinical symptoms to salmonellosis and post-mortems may, therefore, be inaccurate [74].

### 3.6. Stocking Density

Sheep are transported at high stocking densities because of the high cost of travel [75]. The Australian shipping standards [24] require only approximately 0.26 m^2^/head of sheep, dependent on weight. Provision of such little space reduces lying time and the variability of heartbeats, as well as increasing aggression, pushing and stepping, all signs of stress [63]. At such high stocking densities, sheep typically stand with their head facing the floor for long periods [63]. In cattle transported to China, shipments from Australia rarely meet OIE standards for space requirements [48]. 

### 3.7. Unhygienic Environment

On the ship, livestock stand in, and lie on, their own excreta [76]. Unlike in most livestock buildings, there is no way of removing it on a regular basis. Provision of bedding is very limited, but the high ventilation rate onboard ship (required for the high stocking density of animals) normally dries sheep excreta to a friable powder [77]. Cattle excreta has a higher moisture content and remains moist [77]. As the powdered sheep excreta becomes deeper, reaching about 10 cm by the end of the voyage, ammonia accumulates. The powder and high ventilation rate also create dust, which circulates around the ship, potentially causing conjunctivitis [68]. However, in humid conditions, the powdered sheep excreta become a slurry that potentially transmits salmonellosis. Cattle excreta is usually hosed out every few days on the open sea, but not in the vicinity of discharge ports [78]. 

Industry standards [24] only have a specific requirement for bedding on extended long-haul cattle voyages (>31 d, via Cape Hope, Suez Canal, etc.). In this case, the ship should load 7 tonnes or 25 m^3^ for every 1000 m^2^ of cattle pen space, i.e., 7 kg/m^2^. This is spread at the start of the voyage, sometimes after one or two washdowns during the voyage, and then not again until just before arrival [78], i.e., about 30 d later. This represents a depth only 0.6 cm per application [77]. Cattle have a strong preference for, and lie down for longer on, floors with a reasonable amount of bedding [77,79]. With this limited or no bedding provision on most voyages, the floor usually is covered with a slurry of excreta, which contain a high microbial load that potentially contains and transmits pathogens [80]. 

On all other voyages, according to the latest industry standards [24], ‘bedding provisions must be loaded for all voyages and: a) applied in a sufficient quantity that allows pens to be maintained in a manner that ensures the health and welfare of the livestock and … be monitored routinely (at least daily) to ensure consistency and depth is appropriate to mitigate risks to the health or welfare of the livestock.’ As normally no bedding is provided at all [76], it is very unlikely that it is sufficient to contain disease transmission. For example, salmonella organisms are found in high concentrations in slurry in infected cattle herds [81]. They are also found in bedding [81]; therefore, unless sufficient bedding is provided to minimise contact with excreta, the risk of transmission via excreta remains high. 

Transmission of salmonellosis, a serious disease of live export sheep [45], via faeces [82,83,84] and orf (pustular dermatitis) by feeding and drinking in communal troughs, is quite possible [85]. Livestock may defecate into the troughs, which then become potential transmission sites for enteric infections. In assembly depots, only 3 (ad libitum feeding)—5 (rationed feeding) cm of trough per sheep is required by industry standards [24]. On the ship, ‘all livestock must be provided with adequate trough space during the voyage to ensure each animal can meet its daily requirements for feed and water without risk to their health or welfare’ [24]. 

The high ventilation rate on vessels, which has been measured at a mean air speed of 1.4 m/s, but up to 3.5 m/s near ventilation shafts, will potentially transmit respiratory pathogens [70]. Furthermore, as the animals do not move around the pens much, the persistent exposure of animals near ventilation shafts to high air speeds predisposes them to infection by such pathogens [70].

### 3.8. Lighting Sheep Transport Vessels

Livestock dislike going to feed in the dark [86,87], but lights are often left on permanently in live export ships. This may help to reduce disease. Short photoperiods may predispose sheep, whose appetite is influenced by photoperiod, to inappetence [57], potentially exacerbating salmonellosis. 

### 3.9. Disease Risks and Zoonotic Potential

Disease risk onboard ship is associated with farm factors that are not yet well understood. In a study at the end of the last century, one half of the sheep deaths on live export vessels from Australia occurred in sheep from just 14% of farms supplying the sheep [44]. This was a persistent trend over years, related to the relationship between sheep fatness on loading and mortality, rather than the loading of sheep with subclinical disease from certain farms. It is speculated that some farms consistently supply sheep in high condition score, which have low appetites onboard ship, and they then succumb to inappetence. 

### 3.10. Pustular Dermatitis

Pustular dermatitis (scabby mouth or orf) is a common viral disease in Australian sheep for live export. Just over a quarter of a century ago, a survey found that 23% of farms supplying the Australian live sheep export industry were infected with the disease [88]. It causes ulcers around the lips and nostrils and can markedly reduce sheep welfare [89]. Orf is highly contagious to humans, infecting about a quarter of those living or working with sheep [90]. Humans may be infected following contact with infected animals or materials, acquiring itchy, painful blisters and weeping sores. In Australia, the consequences are likely to be less severe than in the challenging medical and hygienic conditions of developing countries to which they are exported. 

The incidence in Australian live export sheep shipments is unknown, but it was recorded at 6% in the Cormo Express, a shipment rejected by Saudi authorities in 2003 because it was above the 5% maximum prevalence [35]. The high concentrations of sheep in the pre-embarkation depot and on the ship provide ideal conditions for transmission of the disease. Vaccination is possible before export, but it is short-lasting and not completely effective in controlling the disease [88,91]. 

### 3.11. Enteric Infections

A major cause of disease in sheep travelling from Australia to the Middle East is salmonellosis, which may be transmitted to humans handling the animals, either on board or on arrival [92]. This gastrointestinal infection is exacerbated by the stress of transport [60], particularly if sheep become inappetent during the journey from Australia to the Middle East [91]. The disease was first reported in live exports of lambs almost 100 years ago, in 1924 [93], when it was largely credited to the inappetence of lambs on board. In humans, salmonellosis is now one of the top ten zoonotic diseases identified by American experts [94] and an extremely common zoonosis. It is often transmitted to people if they do not wash their hands after handling animals or after contact with the animals’ environment, food, water or living quarters. The transmission risk is likely to be increased because of the hand-to-mouth and communal plate eating conventions in the Middle East [95] and other livestock recipient countries in Asia. The hot temperatures in summer aid the bacteria’s survival: optimal temperature for the salmonella organism is 35–43 °C, and it survives up to 46 °C [96]. *S. typhimurium*, the most common serotype affecting sheep on Australia’s live export ships, infects both livestock and humans, and it is of concern that multidrug-resistant serotypes, e.g., DT104, occasionally emerge in Australia [97]. Australia has about 50 reported cases of salmonellosis per 100,000 human population each year, with *S. typhimurium* being the most common serovar [98]. Salmonella has been isolated from 9% of slaughtered feedlot cattle, 20% of sheep faeces and 13% of sheep wool at abattoirs in Australia [99,100]. Other enteric infections also occur: *E. coli* O157 was detected in 5% of faeces (and 3% of fleeces) [100].

It is highly likely that live shipments of sheep and cattle are bringing animals with enteric infections into the Middle East. A study of 518 imported cattle slaughtered at Greater Amman Abattoir, the largest abattoir in Jordan, arriving from S. America and Romania, found that 71% were contaminated with *S. enterica* [101]. Whilst they could have been contaminated whilst awaiting slaughter in Jordan, it is most likely that the cattle were contaminated before arrival. Of even greater concern is that 70% of the infected animals exhibited multidrug resistance and 93% resisted at least one antimicrobial, including resistance to drugs commonly used to treat salmonellosis in humans. 

### 3.12. Respiratory Infections

Pneumonia is another disease commonly seen in exported cattle and sheep [102]. Vaccination is possible, but the wide variety of organisms that infect livestock render adequate protection difficult to achieve [103]. Transport stress is a major predisposing factor in both sheep [104] and cattle [105], as are heat stress, overcrowding, exposure to inclement weather and handling, all of which are associated with live export [104]. Transport stress increases blood neutrophil concentration and serum haptoglobin, typical of a stress response [106]. This may be associated with reduced neutrophil functionality and leads to oxidative stress. The high ammonia concentration in simulated sheep and cattle transport by ship increases pulmonary macrophage activity, indicating inflammation [69], which may predispose the livestock to pneumonia.

The mycoplasmal form in sheep, *Mycoplasma ovipneumoniae*, multiplies rapidly in times of inclement weather [107]. At least one bacterial form, (*Coxiella burnetii*, causing the respiratory infection Q fever, which may progress to pneumonia), is considered ‘extremely hazardous’ to the welfare of cattle, and it is endemic in the northern states of Australia and potentially transmitted to humans [108]. In Australia, workers who are at risk are often vaccinated [109], but the vaccine is not approved in countries to which Australia exports livestock [110], leaving workers there at risk of acquiring the infection. 

Also of concern are coronaviruses, which have a high mutation rate and readily jump the species barrier [111]. Bovine coronaviruses commonly infect the respiratory and intestinal tract of several species and may have infected humans [112]. Coronaviruses were one of top 10 zoonotic diseases identified by American experts just before the world pandemic caused by COVID-19 [94]. In a sample of live export shipments, 13% of cattle had coronavirus infections [113]. Coronaviruses contain a haemagglutinin esterase protein that allows them to bond to multiple cell types [114]. There is an extensive reservoir in wild ruminants, which are under pressure from humans due to deforestation, game farming and overhunting, potentially forcing the virus to mutate in response to novel and stressful environments [63,115]. Australia has large populations of two of these wild ruminants—water buffalo and camels. Camels, which are ruminant livestock like cattle and sheep, appear to have been the intermediate host reservoir species for Middle East respiratory syndrome coronavirus (MERS) [116], since antibodies to the MERS virus have been found in a high proportion of Saudi Arabian camels [117]. This virus posed a particular risk to those working with and killing camels [116], and claimed the lives of almost 1000 people in the early part of the second decade of this millennium [118]. The risk of another coronavirus establishing itself in the human population from imported livestock is significant, especially since cattle commonly have coronavirus infections [119]. Humans travelling with the livestock ships were found to be reservoirs of coronaviruses during the COVID-19 outbreak [120].

With restrictions on Australian sheep exports to Saudi Arabia, the main importer in the Arabian Peninsula for much of the second decade of this millennium, many sheep were sourced for the Middle East market from the horn of Africa, although periodic restrictions also affected this trade [121]. From data derived from livestock slaughtered in this region, it is evident that pneumonia and gastrointestinal parasites are common in livestock in the region. In an Ethiopian abattoir, 43% of lungs were deemed unfit for human consumption, mostly (in 63% of cases) because of pathological lesions caused by pneumonia [122]. Pneumonia is endemic in Middle East sheep flocks and was found in 62% of flocks in a survey of Jordanian sheep and goats [123]. After Rift Valley Fever—a zoonotic infection of livestock—was discovered in Sudan, a temporary ban on imports from Sudan was imposed by Saudi Arabia [124], with reports of several thousand livestock dying of hunger and thirst, both on land and at sea. Normally, Sudanese sheep for export are vaccinated for Rift Valley Fever, but on this occasion, some vaccinated animals appear to have been replaced with unvaccinated ones [124]. 

### 3.13. Conjunctivitis

Contagious ophthalmia, or infectious ovine keratoconjunctivitis, has been measured at between 1 and 2% in a small sample of sheep transported by ship from Australia to the Middle East; 2% if sheep were in a part of the ship with high ammonia concentration [33]. Hot, dry, dusty and windy conditions predispose sheep to the disease [125], all common conditions experienced onboard ships. Many sheep harbour bacteria (e.g., *Moraxella ovis*) in their eyes before export, and the conducive environmental conditions in the pre-export assembly depot and onboard ship trigger outbreaks of the disease [102,126]. Onboard detection and treatment of the disease is difficult, leading to sheep with painful infections and impaired vision, or even blindness, observed in animals being offloaded. *Some* of the causative agents are thought not to be zoonotic [127].

Keratoconjunctivitis is also a problem in exported cattle. Treatment with antibiotics is similarly difficult on board and may be constrained by Export Slaughter Intervals (ESI) imposed by importing countries, e.g., a 90-day ESI for oxytetracycline use in livestock entering Russia, preventing its use in Australian live exports [125]. 

A fungal skin infection—dermatophytosis—in cattle can be induced by the stress of long-distance transport; most common is that caused by the zoonotic pathogen *Trichophyton benhamiae*, which results in *Tinea facei* in humans [43]. 

### 3.14. Non-Communicable Diseases

Although not strictly zoonotic, non-communicable diseases (NCD), such as cardiovascular disease, are associated with consumption of red meat [128], which is the subject of the live export trade from Australia. NCD are perceived to have increased in the Gulf Cooperation Council countries as a result of the growing trade in live animals, increased affluence of the population and resulting overconsumption of meat [129]. These countries have a high proportion of deaths of over 60-year-old people from NCD, particularly obesity and diabetes (33% and 17%, respectively) [130]. 

### 3.15. Controlling Zoonoses That Derive from the Australian Live Export Trade

Vaccination against several zoonoses, e.g., pustular dermatitis, is possible but not completely effective [131]. There is a risk of zoonotic infection in people administering the vaccine, which, together with concerns about vaccine efficacy, means that vaccines are not widely used [132]. However, there are long term benefits of some vaccinations. In Turkish sheep transported long distance, pre-transport vaccination against orf, but not PPR (peste de petits ruminants), improved post-transport immunocompetence [54]. 

Onboard treatment of infectious disease is hampered by difficulties in detection and treatment of affected animals, an unsafe and isolated working environment and importing countries’ restrictions on drug residues [133]. Only about 20% of voyages from Australia have an accredited veterinarian on board [133]. Even if a veterinarian is present, case definitions have not yet been defined, making diagnosis difficult [134].

Loading and unloading represent a significant stress for livestock, which is likely to further reduce resistance to zoonotic disease. For example, cattle in Indonesia are moved between the islands either by being forced into the sea and swimming ashore or being hoisted by a winch around the neck (contrary to national welfare legislation) [135]. Many injuries occur, but these could be avoided, and the stress minimised, with a relatively simple fire engine-type loading ramp for moving cattle on and off the ships [135]. However, currently this is not required or available for loading and unloading [133]. 

There are historical precedents for restriction of the live export trade based on the disease risk. In the 19th century, Britain’s importation of live cattle increased to satisfy the growing affordability of beef in the rapidly industrializing country. However, in 1865, an outbreak of cattle plague (rinderpest) from cattle imported from Europe spread to Britain and caused major cattle losses ([136]. Several centuries earlier, rinderpest had given rise to the measles virus in humans [137,138]. The authorities reacted to the spread of rinderpest to Britain by banning importation of live cattle from areas with known infection and attempting to ensure that British and imported cattle were kept separate, with the latter slaughtered rapidly at the ports after entering Britain [136]. 

Similar problems arose with the importation of live cattle to Britain in 1879 from the USA and Canada, some of which were believed to be infected with pleuropneumonia. Although disputed by some, who claimed it was simply bronchitis, the British government already had in place a legislative order which required all cattle to be slaughtered within 10 days at the port of entry [137]. However, at the time of the 1879 shipment, cattle from the USA and Canada were exempt on the grounds that they were probably free of pleuropneumonia, even though it was known to exist in the eastern states of the USA [139]. Much debate ensued about the diagnosis, and eventually the British government rescinded the American exemption. 

Similar problems emerged in British colonies. In the early 20th century, the Gambia, a British enclave in a region of Africa mostly colonised by France, could not produce enough beef to meet requirements, so it shipped in zebu cattle from neighbouring countries, which were taken upriver to populate the interior [140]. However, the imported zebu cattle had no resistance to trypanosomiasis, unlike Gambian cattle, and only limited resistance to rinderpest. The British authorities in Gambia imposed a movement restriction to try to control the spread of both diseases.

In the 1990s, the Australian trade in sheep to the Middle East experienced many rejections of shipments on account of the high prevalence of pustular dermatitis in the sheep, mostly by the Saudi Arabian authorities, but also by those in Bahrain. The most notorious incident was that of the Cormo Express in 2003, when a Saudi veterinarian found the sheep cargo to have a 6% prevalence of pustular dermatitis, above the limit of 5% [11]. When the sheep were eventually offloaded in Eritrea, after 80 days at sea, mortality had increased to 10%. The Australian government responded to the incident by suspending the trade in live sheep to Saudi Arabia, a ban which lasted until 2005. Although the independent enquiry report by Keniry et al. made many recommendations, none of them addressed the control of pustular dermatitis, even though this was the primary reason for repeated rejection of sheep shipments from Australia [11]. Muslims are specifically forbidden to eat meat that might harm their health, i.e., with zoonotic infections [141]. Hence, animals that are obviously sick—this would include animals with pustular dermatitis—cannot be slaughtered for consumption [142]. Indeed, it is surprising that the Saudis are prepared to even accept shipments with 5% of animals infected, given that the animals have to be handled after disembarking, with potential zoonotic transmission risk. Accurate diagnosis of the early stages of the disease is difficult [143], especially when thousands of sheep are loaded in a short period of time. It presents just as reddened areas in the corner of the mouth and on the muzzle. Given that the disease emerges over 10–14 d after sheep have been exposed to predisposing factors, in this case, high stocking densities and stress, and progresses for 4–8 weeks [144], it is likely that many sheep will be in the early stages during the voyage and may go undiagnosed. Sheep offloaded in Saudi Arabia are taken to markets, where they are often purchased individually or in small numbers and driven to small-scale slaughter operations. This creates a significant risk of zoonotic transmission. 

Since that time, Saudi Arabia refocused its imports on neighbouring Sudan, since Australia required it to participate in its Export Supply Chain Assurance Scheme in 2012 if it wanted continued shipments from Australia [145]. The Saudi authorities refused, since it impinges on Saudi sovereignty over the sheep, but loopholes were eventually found in 2021, which allowed trade between the two countries to resume. Demonstrating their continued commitment to importation of healthy stock, the Saudi authorities had rejected shipments from Sudan when vaccination regulations were breached, and they also suspended the trade when there was an outbreak of Rift Valley Fever in Sudan. 

Cattle exports were also the subject of major trade restrictions worldwide as a result of a new zoonotic disease, bovine spongiform encephalopathy, which first appeared in England in 1986 [146]. Because it is a disease of the nervous tissue and because a connection between the consumption of infected animals with specified high risk materials and a fatal new variant of Creutzfeldt-Jacob disease in humans was strongly suspected, restrictions arising from this zoonotic disease have been severe. The specified risk materials are brain, skull, eyes, trigeminal ganglia, spinal cord, vertebral column (excluding the vertebrae of the tail, the transverse processes of the thoracic and lumbar vertebrae and the wings of the sacrum) and dorsal root ganglia from cattle 30 months of age and older; the distal ileum of the small intestine and the tonsils from all cattle [147]) In many cases, both meat and live animal trade has been suspended upon discovery of any infected animals, although the World Organisation of Animal Health recommend only restrictions on exportation of live ruminant species and their products, removal of the specified risk materials and prohibition of feeding ruminant residues to ruminants [148].

## 4. Conclusions

There is a growing trade in livestock over long distances that presents a significant risk of zoonotic disease transmission. The long-distance trade from Australia, one of the biggest in the world, presents a major risk to importing countries because of the conditions in which they are transported. This creates significant stress over a long period, depleting the animals’ immune system. The stress induced by the long-distance travel, particularly the time on ship from Australia to Asia and the Middle East, is likely to make the animals more susceptible to acquisition of zoonotic infections. The lack of bedding is a serious concern in relation to zoonotic disease transmission, as livestock stand and lie in their own excreta, increasing the risk of transmitting enteric infections. The rest of the world should take note that in the EU, cattle are the subject of the most intense surveillance and disease risk mitigation of any traded livestock, probably because they are perceived to present the greatest risk. All exporting countries, particularly those with a large export (outdegree), should use network analysis to estimate transitivity measures for the livestock trade [2].

The regular emergence of novel zoonoses, such as coronaviruses, and the high prevalence of existing zoonoses emphasizes the growing zoonosis threat posed by the live export industry. It is inevitable that some of those working with livestock, or otherwise in contact with them in recipient countries, will acquire zoonotic disease. It is recommended that there should be better surveillance for novel and existing zoonoses and antimicrobial resistance, and, if necessary, the license to trade suspended for exporters with repeated high levels of zoonotic diseases in their stock. Further study of the ways to mitigate the stress of long-distance transport is warranted, as is government intervention in cases where operators fail to improve standards. Study of the microbiome of exported livestock and its dynamic evolution over the course of the transport processes would potentially identify relationships between commensal organisms and pathogens.

## Data Availability

Not applicable.

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
