# Peer review of "Zoonotic Disease Risks of Live Export of Cattle and Sheep, with a Focus on Australian Shipments to Asia and the Middle East"

_animals, 2022, doi:10.3390/ani12233425_

Round 1
Reviewer 1 Report
this review is very well written and I enjoyed reading it.
I have detect only minimal things which the editorial office can manage:
Line 97. delete double 'during'
Line 113 FAO changed name, now is WOHA, adding double names?
Author Response
Reviewer 1
this review is very well written and I enjoyed reading it.
I have detect only minimal things which the editorial office can manage:
Author’s reply: thank you for your kind words.
Line 97. delete double 'during'
Author’s reply: double during deleted
Line 113 FAO changed name, now is WOHA, adding double names?
Author’s reply: done
Reviewer 2 Report
Dear Editor
The study is a well-designed and well-written work. This review study provides useful information on zoonotic disease risks of live export of cattle and sheep.
.
Comments to the authors
L48: .. recognized in the One
L50: .. live animals are warranted. With most human diseases originating..
L62: …. it is also recognized that there
L65: you could provide more details about the economic impact of Australian livestock sector and exports
L63: … that is enzootic in nature, i.e., stable
L70: .. i.e., countries to which
L80: .. are heat stress, trauma, and respiratory
L114: .. e.g., Livecorp,
L133: … naturally ventilated trucks
L151-153:…. on the destination. Daily management is conducted by the crew, under direction from one or two stock people and sometimes an accompanying veterinarian. A description
L160: .. or as carcasses
L179: … Syria, Egypt, and Turkey
L188: .. Indonesia, the Middle East, and India
L191: .. There are several diseases
L192: .. but also if they are
L209: .. animals can resist colonization
L227: .. during inappetence
L234: .. mobilizing fat tissues
L235: .. fat mobilization
L236: .. sheep currently
L254: .. balance accompanies
L295: .. rate also create dust
L297: … become a slurry
L317: … in slurry and or by feeding
L363: … become inapparent during
L399: .. Mycoplasma ovipneumoniae
L441: .. Hot, dry, dusty, and windy conditions
L406: .. Hot, dry, dusty, and windy conditions
L474: .. stress minimized
L453: .. because of the growing
Author Response
Reviewer 2
The study is a well-designed and well-written work. This review study provides useful information on zoonotic disease risks of live export of cattle and sheep.
.
Comments to the authors
L48: .. recognized in the One
Author’s reply: I’ve used English spelling throughout, not American
L50: .. live animals are warranted. With most human diseases originating..
Author’s reply: changed
L62: …. it is also recognized that there
Author’s reply: I’ve used English spelling throughout, not American
L65: you could provide more details about the economic impact of Australian livestock sector and exports
Author’s reply: changed to: In Australia, the 615,000 cattle exported annually are economically more significant than the almost 500,000 sheep trade, worth Aus $ 1 and 0.085 billion, respectively [9]
L63: … that is enzootic in nature, i.e., stable
Author’s reply: changed
L70: .. i.e., countries to which
Author’s reply: this is not the correct definition of transitivity. See reference 2 for more details.
L80: .. are heat stress, trauma, and respiratory
Author’s reply: In a list, I don’t use commas before the conjunction, as usual in English English.
L114: .. e.g., Livecorp,
Author’s reply: I also don’t use commas after e.g. and i.e. Adding comma is more popular in American English.
L133: … naturally ventilated trucks
Author’s reply: changed
L151-153:…. on the destination. Daily management is conducted by the crew, under direction from one or two stock people and sometimes an accompanying veterinarian. A description
Author’s reply: changed
L160: .. or as carcasses
Author’s reply: English is carcase not carcass
L179: … Syria, Egypt, and Turkey
Author’s reply: In a list, I don’t use commas before the conjunction, as is usual in English English.
L188: .. Indonesia, the Middle East, and India
Author’s reply: In a list, I don’t use commas before the conjunction, as is usual in English English.
L191: .. There are several diseases
Author’s reply: changed
L192: .. but also if they are
Author’s reply: this is a noun phrase in apposition, which needs commas at beginning and end.
L209: .. animals can resist colonization
Author’s reply: no change, colonize is favoured by Americans.
L227: .. during inappetence
Author’s reply: changed
L234: .. mobilizing fat tissues
Author’s reply: no change, mobilize is favoured by Americans.
L235: .. fat mobilization
Author’s reply: no change, mobilize is favoured by Americans.
L236: .. sheep currently
Author’s reply: I don’t think ‘currently’ reflects ‘in autumn after summer drought’ as well as ‘at this time’
L254: .. balance accompanies
Author’s reply: it should be plural as two subjects
L295: .. rate also create dust
Author’s reply: changed
L297: … become a slurry
Author’s reply: changed
L317: … in slurry and or by feeding
Author’s reply: changed to ‘Transmission of salmonellosis, a serious disease of live export sheep [45], via faeces [82,83,84]’
L363: … become inapparent during
Author’s reply: inappetent not inapparent
L399: .. Mycoplasma ovipneumoniae
Author’s reply: changed
L441: .. Hot, dry, dusty, and windy conditions
Author’s reply: In a list, I don’t use commas before the conjunction, as is usual in English English.
L406: .. Hot, dry, dusty, and windy conditions
Author’s reply: In a list, I don’t use commas before the conjunction, as is usual in English English.
L474: .. stress minimized
Author’s reply: minimized favoured in American English
L453: .. because of the growing
Author’s reply: changed to ‘long-distance transport; most common is the zoonotic pathogen’
Reviewer 3 Report
Dear author,
I find this manuscript covering the topic of zoonotic disease risks of live export of cattle and sheep from Australia highly interesting and it is an important contribution to the ongoing debate about this practice. A lot of important knowledge is presented here! Except one principal comment, I have only few and minor comments, and I look forward to see it published. Please see below:
My main comment: I am fully aware of the challenges to document all statements/entences in this manuscript with references, but I think that the current quite high number of statements/sentences that are non-referenced might leave readers with the impression that this is the authors opinion more than evidenced facts. Hence, if possible, I suggest to insert references when possible in the line numbers listed here: 47, 49, 56, 59, 66, 72, 74, 83, 86, 95, 96, 97, 99, 102, 106, 121, 122, 124, 126, 128, 129, 130, 132, 133, 136, 139, 143, 150, 151, 152, 158, 161, 162, 173, 179, 182, 184, 185, 186, 192, 193, 200, 224, 231, 240, 263, 274, 282, 290, 293, 294, 297, 313, 316, 318, 319, 331, 340, 350, 352, 357, 362, 368, 370, 373, 385, 387, 396, 398, 404, 413, 415, 442, 443, 444, 448, 487, 492, 494, 501, 504, 508, 509, 517, 519, 520, 523, 525 twice, 530, 533, 536
If it is not possible, as some of this is probably not described in sources that can be referenced, maybe it can be stated somewhere/somehow?
Minor comments:
L97: Is there a 'during' too much?
L107: There is a big jump in the text here, from background to search strategy. Where is the aim of the study presented?
L109: Should the brackets around 'zoonotic or zoonosis/zoonoses' be deleted?
L113 and elsewhere: Detail, but don't forget that recently OIE changed name to WOAH
L131: Should it be 'stress level'?
L136: Maybe a stupid question, but should it be explained to the readers why this practice is followed - to prevent the animals from lying down?
L143: What is AQIS?
L143: Supposed to? I think I know what you mean, but could this be written more clearly?
L155: 'Loaded onto transport' - is this wording correct?
L158: Can you give an estimate of how long? For some people 4h is long - for others it might need 4 days to be considered long.
L162: Should 'was' be 'is'?
L239: Is the light on all the time? Please explain more. Why?
L239: Should 'believe' be changed to 'perceive' or something like that?
L240: Since this is a suggestion, should it be changed to 'This may make ....'?
L303: Am I miss-reading this sentence, or should it be 'arrival' instead of departure?
L317: I realise that the term 'orf' is explained below. Can it be explained here - the first time it is mentioned?
L330: I don't understand the use of the term 'but' here. Why is it but?
LL331: I don't understand why short photoperiods are mentioned. Isn't the problem the opposite?
L334: Can you explain a bit more why this would be ideally?
L405: Should it be coronaviruses and not coronoviruses?
L446-447: What are they then?
L456: Please explain a bit more what NCD is
Author Response
Reviewer 3
I find this manuscript covering the topic of zoonotic disease risks of live export of cattle and sheep from Australia highly interesting and it is an important contribution to the ongoing debate about this practice. A lot of important knowledge is presented here! Except one principal comment, I have only few and minor comments, and I look forward to see it published. Please see below:
My main comment: I am fully aware of the challenges to document all statements/entences in this manuscript with references, but I think that the current quite high number of statements/sentences that are non-referenced might leave readers with the impression that this is the authors opinion more than evidenced facts. Hence, if possible, I suggest to insert references when possible in the line numbers listed here: 47/, 49/, 56, 59, 66, 72, 74, 83, 86, 95, 96, 97, 99, 102, 106, 121, 122, 124, 126, 128, 129, 130, 132, 133, 136, 139, 143, 150, 151, 152, 158, 161, 162, 173, 179, 182, 184, 185, 186, 192, 193, 200, 224, 231, 240, 263, 274, 282, 290, 293, 294, 297, 313, 316, 318, 319, 331, 340, 350, 352, 357, 362, 368, 370, 373, 385, 387, 396, 398, 404, 413, 415, 442, 443, 444, 448, 487, 492, 494, 501, 504, 508, 509, 517, 519, 520, 523, 525 twice, 530, 533, 536
If it is not possible, as some of this is probably not described in sources that can be referenced, maybe it can be stated somewhere/somehow?
Author’s reply: I have been through the manuscript very thoroughly, introducing approximately 40 new references.
Minor comments:
L97: Is there a 'during' too much?
Author’s reply: yes, one removed.
L107: There is a big jump in the text here, from background to search strategy. Where is the aim of the study presented?
Author’s reply: thank you, the following is added at the start of the paragraph:
The aim of this study was to review literature relevant to zoonotic diseases that might be transmitted during the live export process, in order to assess the risk of transmission of diseases from livestock to humans.
L109: Should the brackets around 'zoonotic or zoonosis/zoonoses' be deleted?
Author’s reply: done
L113 and elsewhere: Detail, but don't forget that recently OIE changed name to WOAH
Author’s reply: included now
L131: Should it be 'stress level'?
Author’s reply: changed, thank you
L136: Maybe a stupid question, but should it be explained to the readers why this practice is followed - to prevent the animals from lying down?
Author’s reply: the following explanation has been added: ‘If they lie down they may be unable to get up again if other animals are standing over them.’
L143: What is AQIS?
Author’s reply: given in full now, Australian Quarantine Inspection Service.
L143: Supposed to? I think I know what you mean, but could this be written more clearly?
Author’s reply: changed to ‘meant to’
L155: 'Loaded onto transport' - is this wording correct?
Author’s reply: changed to ‘loaded into vehicles to transport them to an abattoir’
L158: Can you give an estimate of how long? For some people 4h is long - for others it might need 4 days to be considered long.
Author’s reply: added: ‘several thousand kilometers’ to refer to transport of chickens
L162: Should 'was' be 'is'?
Author’s reply: ‘changed to ‘has been’
L239: Is the light on all the time? Please explain more. Why?
Author’s reply: changed to: ‘The sudden exposure to the continuous light on the ship, left on to reduce accidents,’
L239: Should 'believe' be changed to 'perceive' or something like that?
Author’s reply: changed
L240: Since this is a suggestio
, should it be changed to 'This may make ....'?
Author’s reply: changed to ‘This may make the sheep’…..
L303: Am I miss-reading this sentence, or should it be 'arrival' instead of departure?
Author’s reply: cahnged
L317: I realise that the term 'orf' is explained below. Can it be explained here - the first time it is mentioned?
Author’s reply: (pustular dermatitis) added
L330: I don't understand the use of the term 'but' here. Why is it but?
Author’s reply: it means that it is not a problem because lights are left on.
LL331: I don't understand why short photoperiods are mentioned. Isn't the problem the opposite?
Author’s reply: Some ships will operate short photoperiods, so this considers the impact on the sheep.
L334: Can you explain a bit more why this would be ideally?
Author’s reply: changed to: ‘Ideally, sheep should be exposed to at least 16 hours of light on the ship, but preferably a low lighting level provided during the entire day’
L405: Should it be coronaviruses and not coronoviruses?
Author’s reply: changed
L446-447: What are they then?
Author’s reply: disease restricted to the livestock species
L456: Please explain a bit more what NCD is
Author’s reply: I’ve added: ‘, such as cardiovascular disease,’